# Automatic Time Picking for Weak Seismic Phase in the Strong Noise and Interference Environment: An Hybrid Method Based on Array Similarity

**DOI:** 10.3390/s22249924

**Published:** 2022-12-16

**Authors:** Haofeng Wu, Weiguo Xiao, Haoran Ren

**Affiliations:** 1Ocean College, Zhejiang University, Zhoushan 316021, China; 2Explosion and Seismic Sensing Research Center, Advanced Technology Institute, Zhejiang University, Hangzhou 310027, China; 3Hainan Institute, Zhejiang University, Sanya 572025, China; 4Northwest Institute of Nuclear Technology, Xi’an 710024, China

**Keywords:** automatic arrival time picking, cross-correlation, beamforming, exploration seismology

## Abstract

The extraction of travel-time curve of seismic phase is very important for the subsequent inference of the structural properties of underground media in seismology. In recent years, with the increase in the amount of data, manual processing is facing significant challenges, and automatic signal processing has gradually become the mainstream. According to the similarity of array signals and considering the elimination of outliers, we propose an improved multi-channel cross-correlation method using the $L_{1}$ norm measure to obtain preliminary results, which builds on a new controllable measurement mode. Then, the post-correction step is carried out in combination with the signal gain property of beamforming technique. Based on these two methods, this paper proposes a new scheme of automatic arrival time picking. We apply the scheme to actual data to verify the effects of the two methods step by step. The entire scheme achieves fine results: direct water waves, seismic waves refracted by the crust and seismic waves reflected by the upper mantle are automatically detected. In addition, compared with the two traditional methods, the scheme proposed in this paper has a better overall effect and a reasonable computation cost.

## 1. Introduction

Seismic phase identification and arrival picking are of great significance. It is the basis for subsequent earthquake warning and location [1,2], underground tomography [3,4], etc. With the application of array signal processing technology in the seismic field, array seismology came into being [5]. It enables us to study weak seismic phases under strong background noise that cannot be detected in a single seismogram, extract travel-time curves with higher spatial sampling rate, and obtain more precise tomography results in many local areas. In addition, with the continuous construction of stations around the world and the emergence of new seismic exploration technology based on distributed acoustic sensing [6], massive seismic data processing can no longer be achieved only by manual picking. The demand for an automatic, efficient, and effective method of determining seismic phases is increasing in the field of observational seismology. In fact, array signal processing has applications in many other fields, such as speech recognition [7], underwater acoustic signal processing [8], sensor calibration for augmented reality [9], etc. Arrival picking is often referred to as time delay estimation in the array signal processing community. The solutions to time delay estimation in different fields are likely to be mutually applied.

Since the concept of automatic picking was proposed, most algorithms have been developed for specific datasets or specific problems, such as early warning, real-time positioning, tomography, etc. However, only a few algorithms have been widely used in the field of observational seismology. In these methods, there are mainly three automatic picking principles. The first one is negative feedback logic [10], the seismic signal is declared based on the deviation between the detected signal and the ambient noise. Before the arrival of seismic wave, the station state is 0; upon arrival, the state changes to 1. Therefore, it is often necessary to define a characteristic function to determine the state change point corresponding to the arrival of seismic wave. Common methods include energy transient method [11,12,13], autoregression method [14,15,16], and high-order statistics method [17,18]. The second one is based on the principle of array similarity. In order to improve the picking accuracy, multi-stations or array methods of relative arrival time difference estimation rather than absolute arrival time estimation are proposed. These methods require high waveform coherence and high signal-to-noise ratio of adjacent stations, such as in the case of a low-pass filtered teleseismic waveform. Common methods include multi-channel cross-correlation [19,20,21,22] and adaptive stacking method [23,24]. They quantitatively describe the difference between signals in different ways: cross-correlation is measured by the product of two signals, whereas adaptive stacking is measured by the difference value. The core of adaptive stacking is the idea of iteration. The third is the recently popular machine learning methods [25], which are mainly divided into supervised learning and unsupervised learning. Supervised learning, especially deep learning, is well applied to labelled earthquake seismic datasets. The weights of the neural network are learned from the training set, and satisfactory results are obtained on the test set. For details of algorithm implementation, see SeisBench library [26,27], which integrates six mainstream algorithms. Unsupervised learning works well with unlabelled data, and it is mainly determined by the internal feature similarity of the data. Common methods include clustering [28,29,30,31] and dimension reduction [32].

In this study, we propose a hybrid method to automatically detect weak seismic signals based on the similarity of array waveforms in strong noise and interference environments. First, we improved the traditional multi-channel cross-correlation method for the initial automatic picking of seismic signals. Then, considering strong noise and interference conditions, we considered combining beamforming technology to improve the SNR of the template for post-correction. Finally, we tested the effectiveness of the method step by step on the real data. The travel-time curve of direct water wave Pw was completely extracted, and the tracking of the first break phase could reach 40 km, including seismic wave Pg refracted by the crust and seismic wave PmP reflected by the upper mantle.

## 2. Fundamental and Method Description

### 2.1. Waveform Cross-Correlation Principle

By quantitatively describing the difference between signals, we can find a seismic sequence, that is most similar to a known seismic signal sequence, on a long seismic trace. This can be extended to the extraction of travel-time curve of a certain seismic phase on 2D seismic section (coordinate system composed of time and distance parameters). Therefore, we can mark the arrivals of the same seismic signal on different sensors by selecting a high-quality waveform template on a sensor. Common ways to describe the difference value between signals include calculating the difference value between signals and the product form between signals, while waveform cross-correlation essentially describes the difference between signals in the product form.

We define a seismic time-series containing *N* consecutive samples as $w_{N,\Delta t}\left(t_{0}\right)$, where $t_{0}$ is the time of the first sample, Δt is the sampling interval:(1)$$\begin{array}{c} w_{N,\Delta t}\left(t_{0}\right)={\left[w\left(t_{0}\right),w\left(t_{0}+\Delta t\right),\cdots ,w\left(t_{0}+\left(N-1\right)\cdot \Delta t\right)\right]}^{T}. \end{array}$$

Then, the inner product between $v_{N,\Delta t}\left(t_{v}\right)$ and $w_{N,\Delta t}\left(t_{w}\right)$ is defined by
(2)$$\begin{array}{cc} {\langle v\left(t_{v}\right),w\left(t_{w}\right)\rangle }_{N,\Delta t} & =\langle v_{N,\Delta t}\left(t_{v}\right),w_{N,\Delta t}\left(t_{w}\right)\rangle \\ \; & =\sum_{i=0}^{N-1}v\left(t_{v}+i\Delta t\right)w\left(t_{w}+i\Delta t\right), \end{array}$$
and the fully normalized cross-correlation coefficient by
(3)$$\begin{array}{c} C{\left[v\left(t_{v}\right),w\left(t_{w}\right)\right]}_{N,\Delta t}=\frac{{\langle v\left(t_{v}\right),w\left(t_{w}\right)\rangle }_{N,\Delta t}}{\sqrt{{\langle v\left(t_{v}\right),v\left(t_{v}\right)\rangle }_{N,\Delta t}}\cdot \sqrt{{\langle w\left(t_{w}\right),w\left(t_{w}\right)\rangle }_{N,\Delta t}}}. \end{array}$$

The maximum value of the coefficient *C* is 1, which means that the two time-series are identical. However, due to different environments of sensor deployment and relative difference of source-receiver paths, the similarity of the same signals on different sensors will degrade, i.e., the coefficient C is usually less than 1. The peak value of correlation coefficient may not be very high, but the significance of its peak value relative to the adjacent value is crucial, which is helpful for subsequent time picking. In short, once we select the appropriate waveform template $w_{N,\Delta t}\left(t_{w}\right)$, the target signal $v_{N,\Delta t}\left(t_{v}\right)$ is usually determined by the time corresponding to the maximum peak value of the correlation coefficients.

### 2.2. Arrival Picking with Improved Multi-Channel Cross-Correlation Technique

The signal detection based on cross-correlation method depends on the waveform similarity between signals. In seismic signal acquisition, there are usually two types of station (sensor) deployment: (1) Network consisting of a certain number of relatively sparsely distributed stations (sensors); and (2) dense array of stations (sensors) with small spacing. The stations in the network mainly rely on the single-channel algorithm, and many algorithms based on this principle have been successful. However, the cross-correlation matching algorithm is mainly applied to teleseism, and the time delay of some other near sources at different stations is large, because their propagation paths are different. The difference affected by underground medium is large, which may lead to low similarity of waveform. However, for most of the seismic signals acquired by the array, in addition to the single-channel algorithm, the multi-channel attribute of the array can also be used to pick up the arrival time. Of course, array seismology plays an irreplaceable role in small-scale tomographic structure imaging, such as obtaining more precise underground velocity structure.

#### 2.2.1. Traditional Multi-Channel Cross-Correlation Technique

Traditional multi-channel cross-correlation (MCCC) method has an excellent and incisive understanding of array similarity. For *N* stations, we denote the cross-correlation derived time delay of stations *i* and *j* as $\Delta t_{ij}$. However, due to the existence of noise and interference, discontinuity $r_{ijk}$ will exist, that can be described by the following equation:(4)$$\begin{array}{c} r_{ijk}=\Delta t_{ij}+\Delta t_{jk}-\Delta t_{ik},\;0\leq i,j,k\leq N. \end{array}$$

An $r_{ijk}$ greater than the acceptable error value means unreliable time picking. We can consider establishing the following $C_{N}^{2}$ equations for more robust time delay estimation:(5)$$\begin{array}{c} t_{i}-t_{j}=\Delta t_{ij},\;i=1,2,\cdots ,N-1,\;j=i+1,i+2,\cdots ,N. \end{array}$$

An additional constraint equation is considered to establish a non-singular over-determined linear system of equations:(6)$$\begin{array}{c} \sum_{i=1}^{N}t_{i}=0. \end{array}$$

The elimination of time picking inconsistency can be attributed to solving an over-determined linear system of equations. Take N=4 as an example, its matrix form is:(7)$$\begin{array}{c} A^{\prime }t=\Delta^{\prime }, \end{array}$$
where
$$A^{\prime }=\left[\begin{array}{cccc} 1 & -1 & 0 & 0\\ 1 & 0 & -1 & 0\\ 1 & 0 & 0 & -1\\ 0 & 1 & -1 & 0\\ 0 & 1 & 0 & -1\\ 0 & 0 & 1 & -1\\ 1 & 1 & 1 & 1 \end{array}\right],t=\left[\begin{array}{c} t_{1}\\ t_{2}\\ t_{3}\\ t_{4} \end{array}\right],\Delta^{\prime }=\left[\begin{array}{c} \Delta t_{12}\\ \Delta t_{13}\\ \Delta t_{14}\\ \Delta t_{23}\\ \Delta t_{24}\\ \Delta t_{34}\\ 0 \end{array}\right].$$

To change t into the same relative delay quantity as $\Delta^{\prime }$, we obtain a new matrix representation:(8)$$\begin{array}{c} A\tau =\Delta , \end{array}$$
where
$$A=\left[\begin{array}{ccc} 1 & 0 & \;\;0\\ 0 & 1 & 0\\ 0 & 0 & 1\\ -1 & 1 & 0\\ -1 & 0 & 1\\ 0 & -1 & 1 \end{array}\right],\tau =\left[\begin{array}{c} \tau_{12}\\ \tau_{13}\\ \tau_{14} \end{array}\right],\Delta =\left[\begin{array}{c} \Delta t_{12}\\ \Delta t_{13}\\ \Delta t_{14}\\ \Delta t_{23}\\ \Delta t_{24}\\ \Delta t_{34} \end{array}\right].$$

Equation (Equation 8) has the weighted least square solution in the form of:(9)$$\tau ={\left(A^{T}\mathrm{WA}\right)}^{-1}A^{T}W\Delta ,$$
where W is a weight matrix. If W=I, τ becomes an unweighted solution. There are mainly two weighting schemes: (1) the correlation value corresponding to the measured time delay; and (2) a weighting function that uses the deviation value $r_{ij}$ of unweighted estimates. Finally, the estimation of the timing uncertainty at each trace is defined as the standard deviation of the deviation:(10)$$\begin{array}{c} \sigma_{i}^{r}=\sqrt{\frac{1}{N-2}\cdot \sum_{j=1,j\neq i}^{N}r_{ji}^{2}}. \end{array}$$

#### 2.2.2. Improved Multi-Channel Cross-Correlation Technique

We can solve Equation (Equation 8) from the perspective of optimization. The least square solution of MCCC corresponds to
(11)$$\begin{array}{c} \min\;∥{W\cdot (A\tau -\Delta )∥}_{2} \end{array}$$

Actually, the least square solution, namely, the $L_{2}$-norm residual minimization, is not robust to large outliers [33]. A better option to consider is the $L_{1}$-norm minimization of residuals:(12)$$\begin{array}{c} \min\;∥{W\cdot (A\tau -\Delta )∥}_{1} \end{array}$$

This is because the $L_{2}$-norm residual minimization averages the estimated arrival times of (N−1) stations, and the $L_{1}$-norm residual minimization takes the median of the estimated arrival times of (N−1) stations. As seen in Figure 1, quantitatively describing the concentration center and dispersion degree of data based on the median can appropriately exclude outliers. In addition, we suggest using interquartile range instead of variance as the uncertainty estimate.

Then we can consider the measurement mode of $\Delta t_{ij}$. Generally, we select the target signal on the first trace, and then match the same signal on the subsequent trace to obtain the base time of the first line, as shown in the measurement mode 1 in Figure 2. On this basis, we further compute the elements on the second line until all elements in the upper half of the matrix are derived. However, the delay search range of this mode is very large, which is completely determined by the first line, and it is very easy to be disturbed from other signals. Here, we propose a new measurement mode, as shown in the measurement mode 2 in Figure 2. The base time we first compute is the elements in red box and we associate it with q=1. The value of *q* measures the size of the delay search range and the redundancy of the array information we use. The smaller the q, the more effectively we can suppress the interference of other interfering signals. Therefore, q=1 guarantees the reliability of the base time. Of course, the choice of *q* has a trade-off between using more redundant information and suppressing other unwanted signals.

Based on the above considerations, we propose an improved MCCC, which can be summarized as the standard form of convex optimization problems:(13)$$\begin{array}{cc} \min & \;∥W_{q_{m}}\cdot {\left(A_{q_{m}}\tau -\Delta_{q_{m}}\right)∥}_{1}\\ s.t. & \;0⪯\tau ⪯\left(N-1\right)\cdot t_{m}, \end{array}$$
where $\Delta_{q_{m}}$ is the column vector consisting of all the elements satisfying $q\leq q_{m}\left(\leq N-1\right)$ in the measurement mode 2, $A_{q_{m}}$ is a new matrix consisting of the row vectors of A corresponding to the elements in $\Delta_{q_{m}}$, τ is the time delay estimator to be solved, $W_{q_{m}}$ is a diagonal matrix, whose diagonal elements are composed of the diagonal elements of W corresponding to the elements in $\Delta_{q_{m}}$.

### 2.3. Iterative Post-Correction Using Beamforming Technique

A very powerful technique in array signal processing is beamforming, as shown in Figure 3. From the real monitored seismic data, coherent signals on different sensors in the array are often surrounded by background noise and incoherent signals. In some extreme cases, the amplitude of weak coherent signal is close to that of background noise. In this case, it is not enough to use only the signal on a single sensor as a template, but beamforming technique can use weak signals on multiple sensors to stack and generate waveform with higher signal-to-noise ratio, which is more powerful for matching target signals. Here, for the selection of weighted stack type, we recommend phase-weighted stack (PWS) [34], whose core idea is to strengthen coherent stacks and suppress incoherent stacks more strongly by virtue of the extremely sensitive coherence of signals in the instantaneous phase. The formula can be described by:(14)$$\begin{array}{c} \mathrm{PWS}\left(t\right)={\left|\sum_{k=1}^{N}\exp\left[i\Phi_{k}\left(t\right)\right]\right|}^{I}\cdot \left(\frac{1}{N}\sum_{j=1}^{N}w_{k}\left(t\right)\right) \end{array}$$
where $w_{k}\left(t\right)$ is the amplitude value of the signal on trace *k* at time *t*, $\Phi_{k}\left(t\right)$ is the instantaneous phase of the signal on trace *k* at time *t*, *I* is the exponent value of power operation.

After obtaining the base time utilizing improved MCCC method, we can use the beamforming technique to generate a beam, then cross-correlate each trace with the beam within a small pre-defined range, and adjust the arrival time of target signal on each sensor by the lag corresponding to peak value of correlation coefficient. Next, we stack each trace to form a updated beam and repeat the above cross-correlation correction procedure to obtain a new beam. The iteration process can be stopped until a convergence condition is satisfied. Therefore, the flowchart of automatic picking scheme based on array similarity is finally shown in Figure 4.

## 3. Results

We mainly use a real marine seismic exploration dataset to conduct a step-by-step test on the proposed method. The data used in the tests come from the wide-angle seismic experiment in the central basin of the South China Sea [35]. The wide-angle seismic profile (OBS2014-ZN) is east–west along the latitude of 14.5° N. The project was completed in July 2014, using 12 OBS at intervals of 10 km. The total length of blasting inspection is 210 km, which is 50 km longer than the two terminal stations. The seismic source provided by Shiyan-2 marine scientific research vessel was an air gun array of 4 × 24.5 L, which was fired every 120 s. The average shooting interval was approximately 280 m. GPS was used to determine the location and time during OBS deployment and recovery, ship navigation and air shooting. The sampling rate of the four-component OBS data is 250 Hz. It must be pointed out that we only used the data recorded by the hydrophone component of OBS09, where only the 135 seismic traces were chosen. The seismic profile composed of 135 traces is shown in Figure 5, where 2–20 Hz band-pass filtering is used.

In order to increase the understanding of the law of wave propagation contained in the data, we can use a two-dimensional transversely uniform initial velocity model in Table 1 to obtain the theoretical travel-time curve, as shown in Figure 6. From Figure 6b, we can recognize that the three seismic phases are direct water wave Pw, seismic wave Pg refracted by the crust and seismic wave PmP reflected by the upper mantle. There is also a noticeable phenomenon in Figure 6a that the first break phase varies with the epicenter distance. When the distance is very small, Pw arrives first; as the distance increases, Pg exceeds Pw; finally, PmP slightly exceeds Pg. Thus, phase crossovers occur twice.

### 3.1. Real Data Test I: Improved Multi-Channel Cross-Correlation Technique

First of all, we pick the arrival time of the direct water wave Pw. The signal-to-noise ratio of Pw is relatively high. With the increase in epicenter distance, the amplitude attenuation of the direct water wave is relatively small, and the spectrum distribution is relatively stable. Then, 2 Hz high pass filtering is applied to the raw data. Figure 7 demonstrates the actual results of MCCC and improved MCCC for Pw tracing, where we use the measured information corresponding to 1≤q≤20. It is obvious that improved MCCC is superior to MCCC on Pw tracing. The timing uncertainty of MCCC and improved MCCC are given in Figure 8. After careful observation, it is found that improved MCCC is a better method, and interquartile range is more suitable for uncertainty estimation. The interquartile range on each trace is within a sampling interval (0.004 s).

Next, we turn to weak first arrival picking. It should be pointed out that, due to the attenuation of the amplitude of the first break, the amplitude of the first break is similar to that of the surrounding signal at a long distance. The measured information other than q=1 cannot be used, because they will not bring about time correction, but will make the base travel time curve strongly oscillate. For a relatively reliable base time, we set $q_{m}=1$. Here we make a hypothesis, that is, select the time delay corresponding to the first cross-correlation coefficient peak value to determine the first arrival signal. The velocity of seismic wave is generally more than 3 km/s, and even more than 8 km/s in the upper mantle. The offset between adjacent traces here is about 300 m, so it can be calculated that the time delay of the first arrival in the adjacent traces is less than 0.1 s, which is less than the half period of the first arrival phase (about 0.2∼0.3 s). Therefore, this assumption is reasonable. We can set a small search range, e.g., 0.15 s. This can not only contain the first break, but also prevent cycle-skipping. For robust first arrival picking, we limit the pass-band of band-pass filtering to 2∼10 Hz to improve the similarity between waveforms. The picked first break time (Base time) is marked in Figure 9a. Globally, base times are located near the first breaks, and most of them are accurate. In fact, if the detected signal on one trace is poor, it will affect all subsequent picks. However, we found that the local consistency performance is well maintained. For example, at the offset ranging from −25 to −28 km, the base times fall on the wave crest, while nearby picks fall on the vibration starting point. The more obvious alignment results can be seen in Figure 10a.

### 3.2. Real Data Test II: Hybrid Method Considering Beamforming Technique

#### 3.2.1. The Hybrid Method Proposed in the Paper

The base time, obtained from the real data test I, can be used continuously in combination with the beamforming technique for post-correction. First, we choose I=1.5 as the specific type of all stacks. It has relatively fine signal enhancement effect. The waveform obtained by phase-weighted stacking according to base time is shown in Figure 10a. We select the signal template on the stack trace, and set the time range of the left and right relative translation of the template to −0.15∼0.15 s, and then perform cross-correlation correction operation for each trace in turn, a new stack waveform can be computed by the updated time delay. Four iterations were implemented. The results after these four iterations are shown in Figure 9. The overall results are quite good. From Figure 9a, it can be found that the results after iteration are significantly improved in the range of −25∼−28 km, and the picked arrivals are obviously corrected from the wave crest to the starting point. Figure 9b demonstrates the difference of results after different iterations. The results after each iteration are very similar, which means that the correction results are basically convergent after one iteration. The stability of post-correction on the basis of base time is extremely high. In order to have an intuitive understanding of the accuracy of the picked arrivals before and after the iteration, we flatten the picked travel time curve to show the seismograms, which intercepts the data from 2 s before arrivals to 3 s after arrivals, as shown in Figure 10. The increase in the maximum amplitude of stacked waveforms in Figure 10c,d proves that better first break alignment is achieved after iteration, so the waveform coherence in the array is further improved.

Figure 6a explains that the first break includes two stages, the first stage is Pg and the second stage is PmP. According to Figure 9a, we roughly determine that the first break within 18 km is Pg and that beyond 23 km is PmP. We compare the results of the two stages before and after iteration, as shown in Figure 11. The results show that good and significant stacked waveforms can be obtained based on the arrivals within array derived by the hybrid method, and it can also be effective in the case of strong background noise surrounding the first break signal. Apart from this, we can calculate the similarity matrix between signals to evaluate the quality of the picked signals. However, considering the low-SNR feature of the signals, we stack five adjacent traces to obtain a beam with a higher SNR. Therefore, 135 traces can form 27 beams from near to far according to the epicenter distance. It should be noted that the similarity of inter-beam $S_{\mathrm{inter}}\left(m,n\right)$ is used here as the quality assessment measure, as defined by Equation (Equation 15). For these beams, the calculated similarity based evaluation results are shown in Figure 12. It can be seen from Figure 12a that the signals before iteration are mainly divided into two clusters: orange branches (1∼11,18) and green branches (14,15,19∼25). Figure 12b shows that the overall array dissimilarity decreases after post-correction, and the main cluster number is also reduced to 1, that is, the set composed of yellow branches.
(15)$$\begin{array}{c} S_{\mathrm{inter}}\left(m,n\right)=\frac{\sum_{j=1}^{5}{\left[w_{i,m}\left(t_{i,m}+j\cdot \Delta t\right)+w_{i,n}\left(t_{i,n}+j\cdot \Delta t\right)\right]}^{2}}{2\cdot \sum_{j=1}^{5}\left[w_{i,m}^{2}\left(t_{i,m}+j\cdot \Delta t\right)+w_{i,n}^{2}\left(t_{i,n}+j\cdot \Delta t\right)\right]}. \end{array}$$

#### 3.2.2. Comparison with Traditional Single-Channel Algorithms

The single-channel algorithm is simple and effective, and has achieved great success in the single-station automatic detection algorithm. We can also apply the negative decision logic (NDL) algorithm to operate on each seismogram that constitutes a two-dimensional section. Here, we directly utilize the PhasePApy library [36]. Two types of NDL algorithms are selected, including multi-band long and short time window (STA/LTA) method and AR-AIC method. Considering the detection principle of the NDL algorithm, we use 2∼30 Hz for band-pass filtering to ensure that the characteristic (energy) of seismic wave changes significantly before and after arrival. The results obtained by the NDL algorithms are shown in Figure 13. Obviously, STA/LTA method has a poor effect on picking weak first breaks. It conforms to common sense that STA/LTA method works well at high SNR, but usually fails at low SNR. Actually, AR-AIC method is a more advanced NDL method, especially in microseismic detection. The results also show that it has a surprising effect as a single-channel algorithm that a majority of first breaks are accurately extracted. In addition, we show the running times of the three methods in Table 2, to compare the computational cost. In contrast, the method proposed in this paper can effectively control the computational cost while ensuring good performance.

## 4. Conclusions and Discussions

We propose a new arrival picking scheme for seismic phase, which combines improved multi-channel cross-correlation and beamforming technique. This is especially suitable for the ensemble of seismograms detected under the small-scale array spacing structure. We have tested our method step by step on 135 real seismic traces. The results show that despite the existence of strong noise and interference, the scheme still obtains good results, and the longest distance of first break picking can reach 40 km, which is almost close to the limit of manual identification. We also found an interesting phenomenon from Figure 13, that is, there is a phase cross-over at about −25∼−26 km. The type of the first break phase will change from Pg to PmP. This verifies the arrival law of the first break wave derived from the theoretical velocity model in Figure 6a. Finally, We compare the method proposed in this paper with two common automatic single-channel first break picking algorithms (multi-band STA/LTA and AR-AIC method), and the results show that the method proposed in this paper has the best global results, and its calculation cost is relatively reasonable and comparable with AR-AIC method.

We have noticed that the deep learning method has achieved great success in earthquake seismology because of the huge amount of artificially analyzed seismic data. The data-driven method may not be easy to directly migrate to exploration seismic data in smaller areas. Because the training set lacks seismic data from this region, it will reduce the generalization of deep learning. However, the method proposed in this paper can provide well-labeled seismic data for the depth neural network. In the future, we can try to apply deep learning method to the result of this work, which has a good prospect of intelligent application.

The proposed method in this paper has great potential to be applied to many other scientific and engineering problems involving time delay estimation in the field of array signal processing, such as radar ranging, wireless location, and sonar direction finding. 

## Figures and Tables

**Figure 1 sensors-22-09924-f001:**
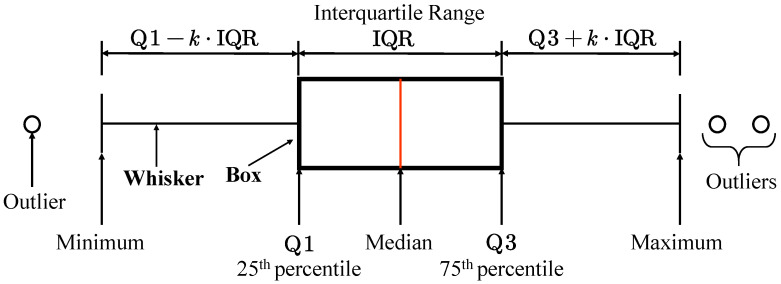
Schematic presentation of Box and Whisker plot. The value of *k* in the figure is 1.5, if you use the function matplotlib.pyplot.boxplot or pandas.DataFrame.boxplot in Python to generate it.

**Figure 2 sensors-22-09924-f002:**
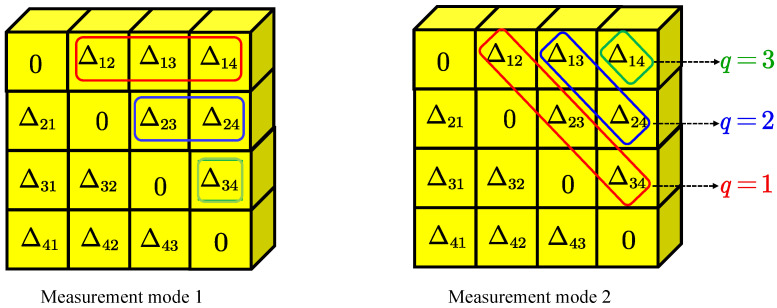
Two different measurement modes.

**Figure 3 sensors-22-09924-f003:**
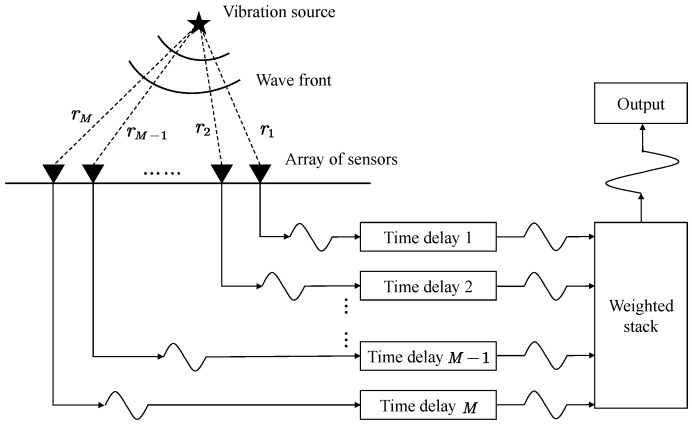
Schematic diagram of the beamforming technique. By compensating the arrival time difference of the same signal on different sensors, and then weighting and summing all the signals, a more significant signal can be obtained.

**Figure 4 sensors-22-09924-f004:**
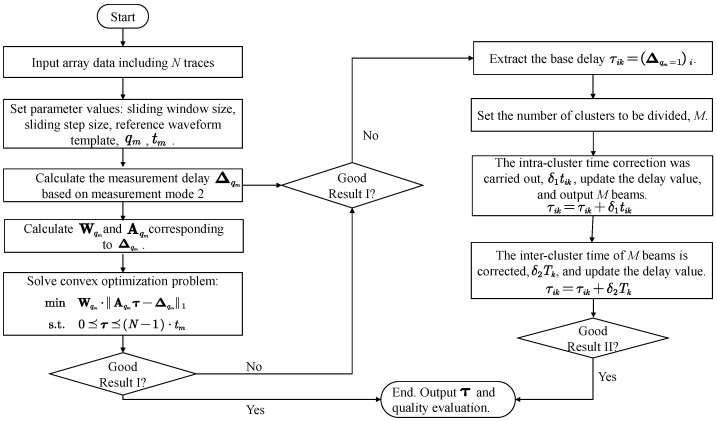
Flowchart of the hybrid method proposed in this paper.

**Figure 5 sensors-22-09924-f005:**
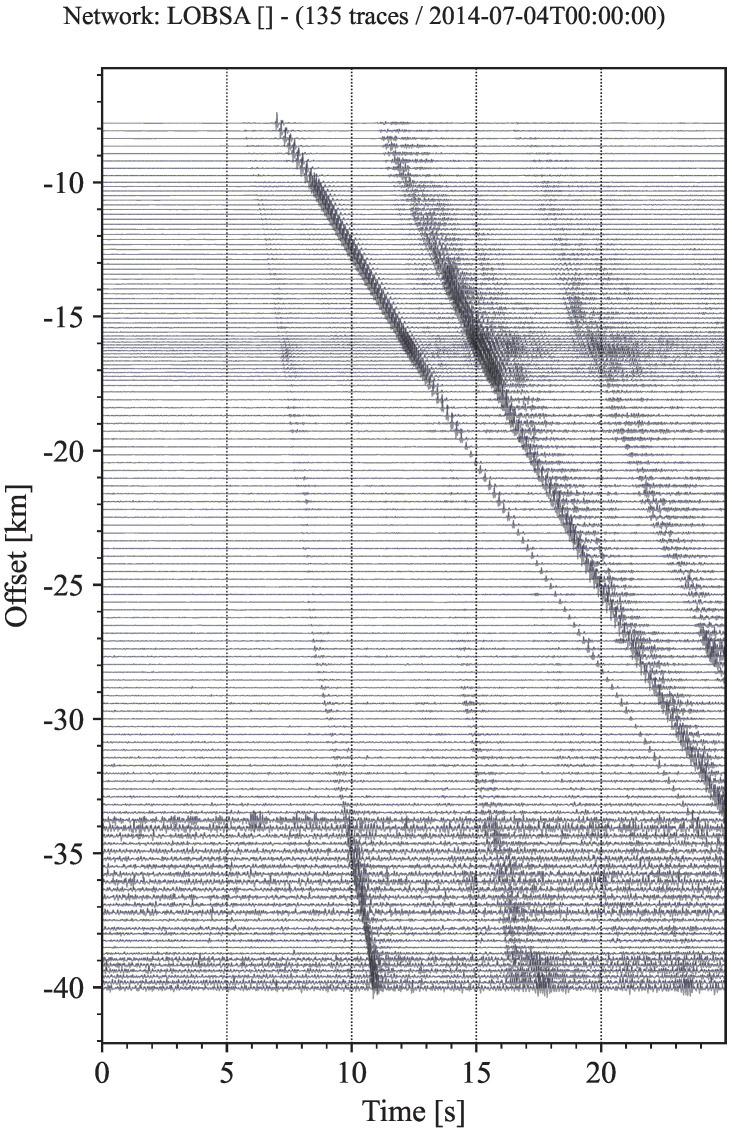
Seismic section, including 135 traces. All traces are normalized separately to their respective absolute maximum.

**Figure 6 sensors-22-09924-f006:**
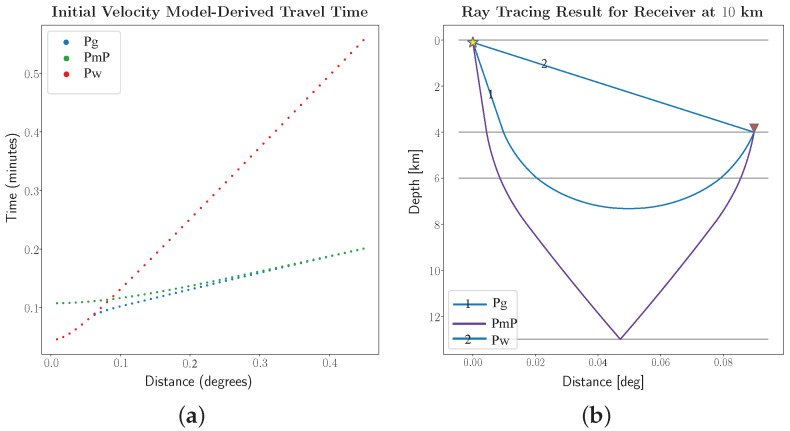
Calculation of theoretical travel time curve. (**a**) Travel time curves of three main phases, with epicentral distance ranging from 1 to 50 km ($1^{\circ }\approx 111.32$ km). (**b**) The theoretical propagation paths of the three main seismic phases in (**a**) received by the sensor at 10 km. The star represents the source of the explosion, and the inverted triangle represents the receiver.

**Figure 7 sensors-22-09924-f007:**
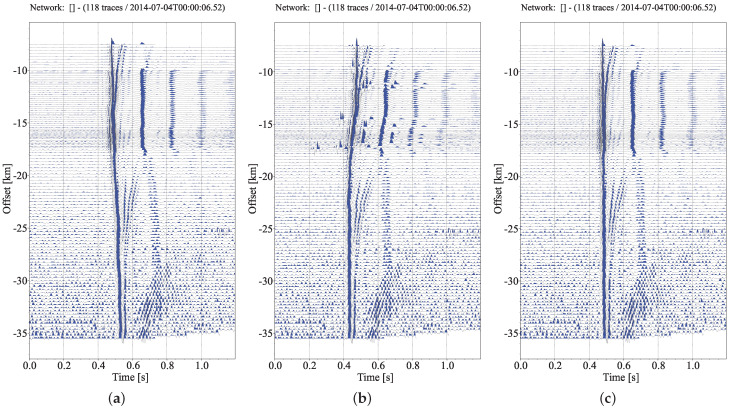
Seismic phase Pw alignment corrected by moveouts: (**a**) base time delay Lag1 obtained by measurement mode 2; (**b**) Lag2 estimated by MCCC; and (**c**) Lag3 estimated by improved MCCC ($q_{m}=20$).

**Figure 8 sensors-22-09924-f008:**
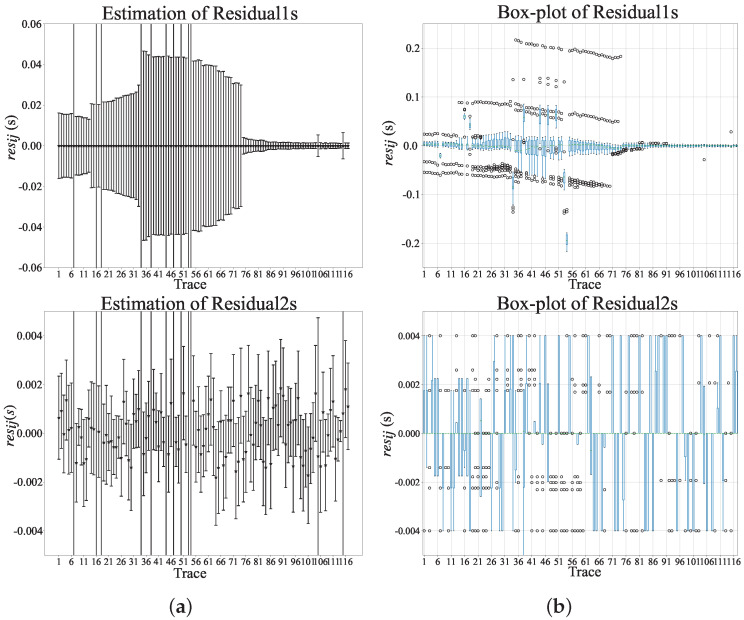
Assessment of results obtained by MCCC and improved MCCC: (**a**) mean and variance, and (**b**) median and interquartile range based on residual1s (Lag2−Lag1) and residual2s (Lag3−Lag1).

**Figure 9 sensors-22-09924-f009:**
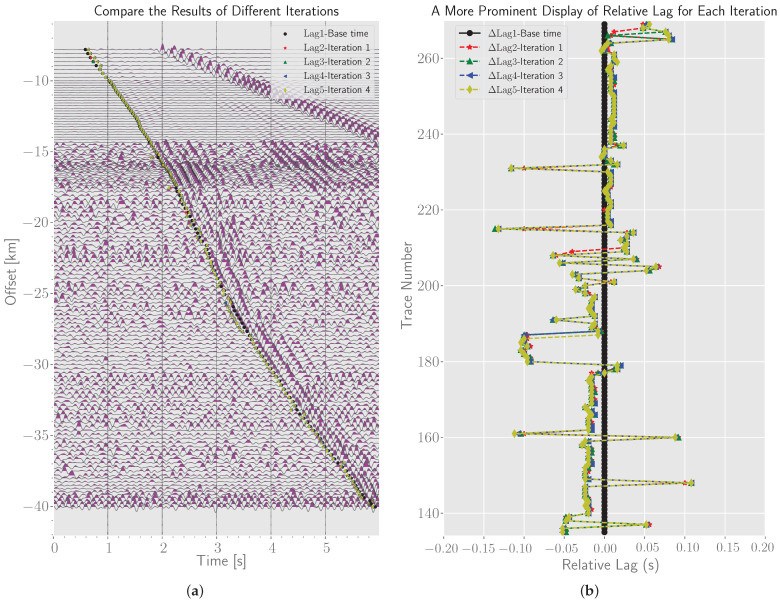
The results of first-arrival seismic phase identification using a hybrid method. (**a**) Auto-picked travel time points marked on 2D seismic section. (**b**) Magnified display of the differences between the results after four iterations and the base result.

**Figure 10 sensors-22-09924-f010:**
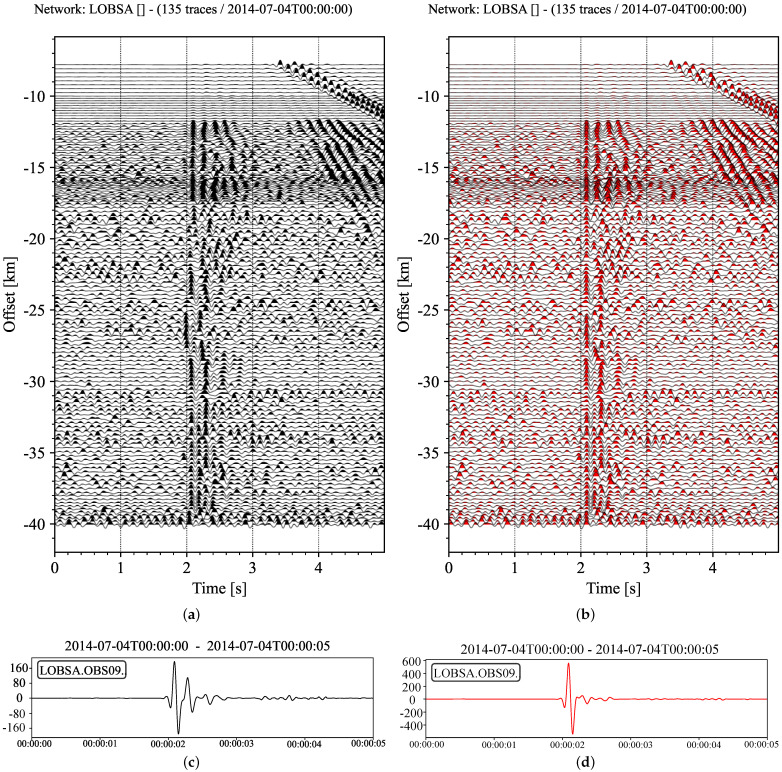
First-arrival seismic phase alignment corrected by moveouts: (**a**) base time; (**b**) the time obtained by post-correction after one iteration. (**c**,**d**) are the phase weighted stacking results of all seismograms in (**a**,**b**), respectively.

**Figure 11 sensors-22-09924-f011:**
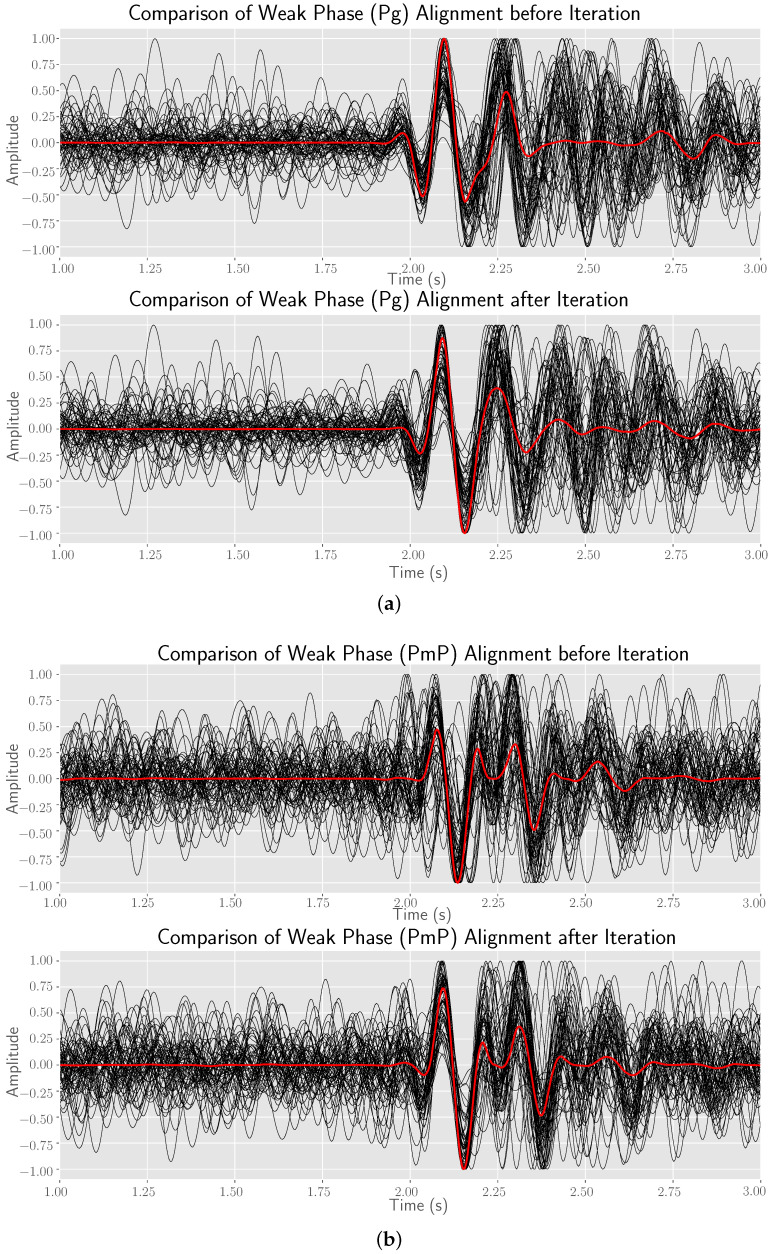
The stacked waveforms of all seismograms within different stages before and after iteration. (**a**) The first stage corresponds to the traces 213∼270; (**b**) the second stage corresponds to the traces 135∼196. The black solid line is the segment of each seismogram (the starting point is 2 s before arrival and the ending point is 3 s after arrival), and the red solid line is the stacked waveform of all seismograms.

**Figure 12 sensors-22-09924-f012:**
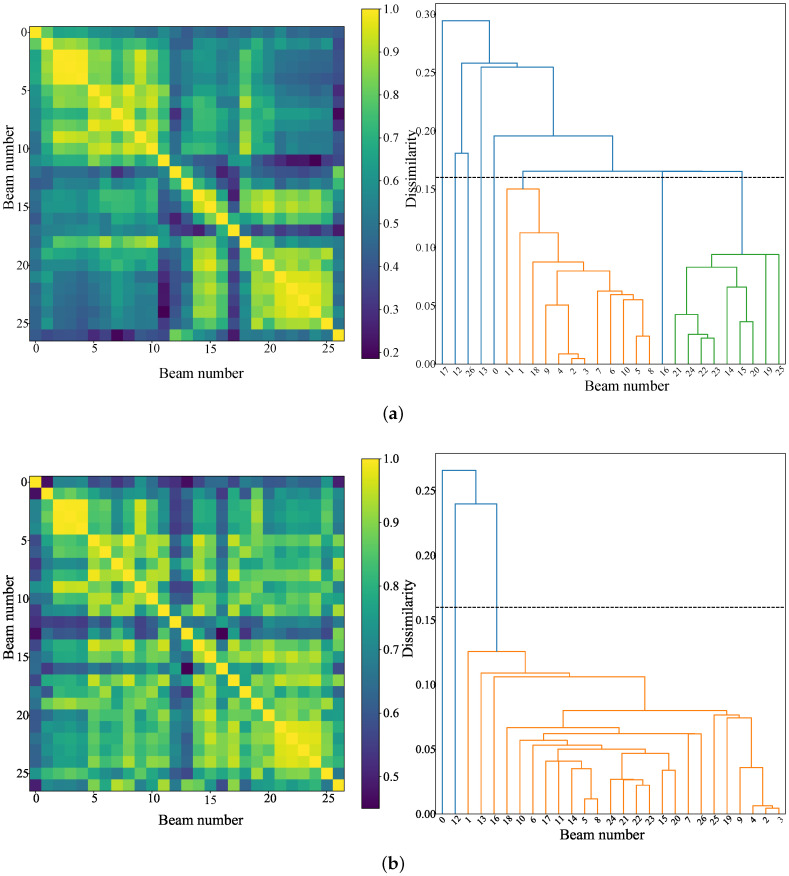
The similarity matrices and resulting dendrograms calculated from 27 stacked beams: (**a**) drawn before iteration and (**b**) drawn after iteration. The dissimilarity threshold is set to 0.16, and the branches with different colors represent different hierarchical clusters.

**Figure 13 sensors-22-09924-f013:**
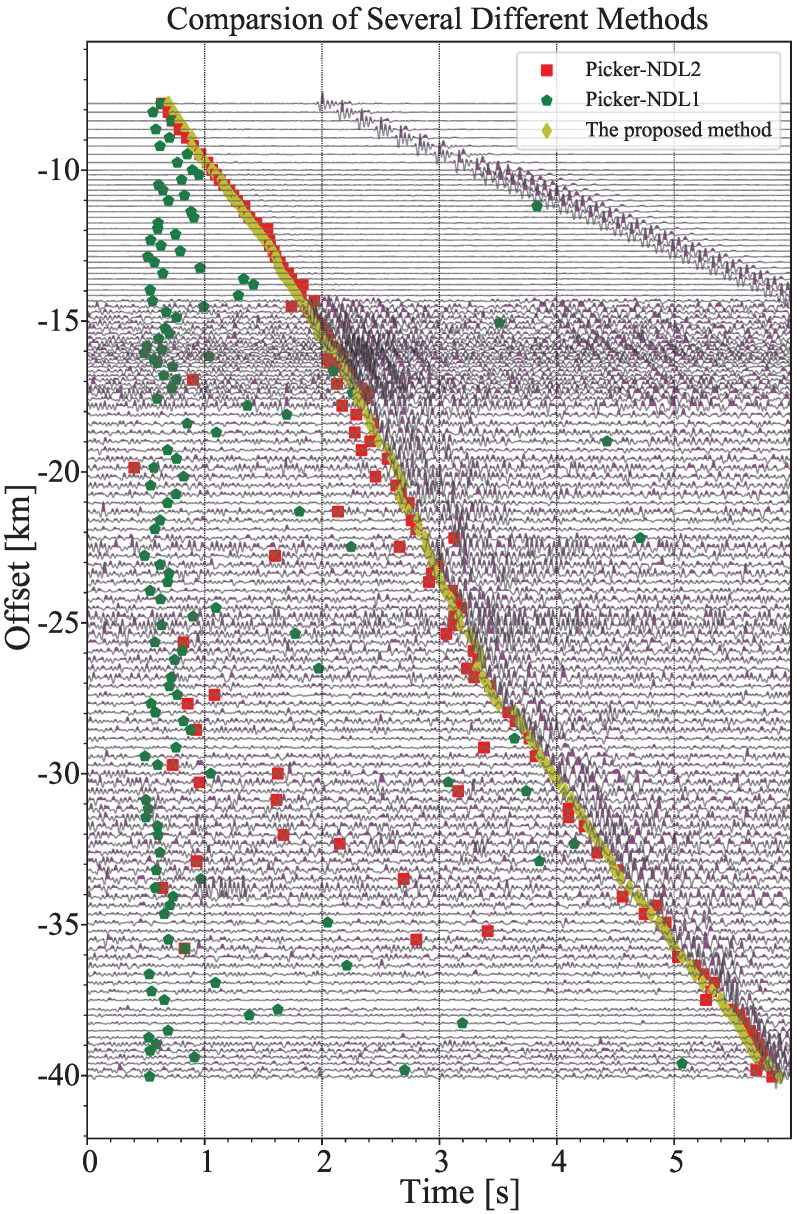
Comparison of the results of the proposed method and the typical NDL algorithms. Picker-NDL1 and Picker-NDL2 denote multi-band STA/LTA and AR-AIC method, respectively. The arrivals denoted by the proposed method are the same as arrivals Lag5 in Figure 9a. All seismograms displayed are filtered by 2∼30 Hz bandpass.

**Table 1 sensors-22-09924-t001:** Initial two-dimensional velocity model for travel time forward modeling.

Layer Type	Depth	$V_{\mathrm{top}}$	$V_{\mathrm{bottom}}$
Seawater layer	0∼4 km	1.5 km/s	1.5 km/s
Sedimentary layer	4∼6 km	1.8 km/s	3.8 km/s
Upper crustal layer	6∼8 km	4.8 km/s	6.4 km/s
Lower crustal layer	8∼13 km	6.4 km/s	7.0 km/s
Upper mantle layer	13∼20 km	8.0 km/s	8.2 km/s

**Table 2 sensors-22-09924-t002:** Comparison of the computational cost for the three different methods.

Method	Multi-Band STA/LTA	AR-AIC	The Proposed Method
Time	1.8 s	13.5 s	16.4 s

## Data Availability

Respective programs and data for reproducibility will be available soon in https://github.com/HaofengWu20/An-Hybrid-Method-Based-on-Array-Similarity-Sensors-2022- (accessed on 13 December 2022).

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
