# Peer review of "Automatic Time Picking for Weak Seismic Phase in the Strong Noise and Interference Environment: An Hybrid Method Based on Array Similarity"

_sensors, 2022, doi:10.3390/s22249924_

Round 1

Author Response

Thank you very much for your valuable comments. Below is our response.

Reviewer 2 Report

This manuscript is one of the very well-written ones I read recently. The authors present a new automatic first-arrival picking method that compares favorably to conventional ones despite its computational cost.

Author Response

Thank you very much for your valuable comments.

Reviewer 3 Report

The manuscript describes an improvement of the automatic picking procedure using an improved multichannel cross correlation method.

The increasing amount of data in the seismological field certainly requires an automation and improvement of the picking process. The subject of the manuscript, to my knowledge, is original and deserves to be published in Sensors.

Minor revisions:

(1) The procedure is well described I was able to follow the discussion. However, in the seismological field, sometimes, analysis are often made not from the waveform directly but from the envelope function obtained through the Hilbert transform of the signal. In the opinion of the authors, could using a transformed signal make any improvement to the picking procedure?

(2) I believe that for the reproducibility of the results it is really important to make available the code that has been developed. I don't see any information in the Data Availability Statement. If the code is still in a preliminary form, it would still be nice to create a repository on which to insert the developments. In any case, some more information on the technical part would be welcome.

Author Response

(The authors gave the same response as above.)
